# Interfacial Interaction Enhanced Rheological Behavior in PAM/CTAC/Salt Aqueous Solution—A Coarse-Grained Molecular Dynamics Study

**DOI:** 10.3390/polym12020265

**Published:** 2020-01-25

**Authors:** Dongjie Liu, Yong Li, Fei Liu, Wenjing Zhou, Ansu Sun, Xiaoteng Liu, Fei Chen, Ben Bin Xu, Jinjia Wei

**Affiliations:** 1School of Chemical Engineering and Technology, Xi’an Jiaotong University, Xi’an 710049, China; jun080511@stu.xjtu.edu.cn (D.L.); wj.zhou@mail.xjtu.edu.cn (W.Z.); 2State Key Laboratory of Multiphase Flow in Power Engineering, Xi’an Jiaotong University, Xi’an 710049, China; bestceo@xjtu.edu.cn; 3Drilling and Production Engineering Research Institute, Chuanqing Drilling and Exploration Engineering Company Ltd., CNPC, Xi’an 710018, China; liyong_gcy@cnpc.com.cn; 4Mechanical and Construction Engineering, Faculty of Engineering and Environment, Northumbria University, Newcastle upon Tyne, NE1 8ST, UK; ansu.sun@northumbria.ac.uk (A.S.); terence.liu@northumbria.ac.uk (X.L.)

**Keywords:** rheology, shear viscosity, shear rates, molecular dynamic

## Abstract

Interfacial interactions within a multi-phase polymer solution play critical roles in processing control and mass transportation in chemical engineering. However, the understandings of these roles remain unexplored due to the complexity of the system. In this study, we used an efficient analytical method—a nonequilibrium molecular dynamics (NEMD) simulation—to unveil the molecular interactions and rheology of a multiphase solution containing cetyltrimethyl ammonium chloride (CTAC), polyacrylamide (PAM), and sodium salicylate (NaSal). The associated macroscopic rheological characteristics and shear viscosity of the polymer/surfactant solution were investigated, where the computational results agreed well with the experimental data. The relation between the characteristic time and shear rate was consistent with the power law. By simulating the shear viscosity of the polymer/surfactant solution, we found that the phase transition of micelles within the mixture led to a non-monotonic increase in the viscosity of the mixed solution with the increase in concentration of CTAC or PAM. We expect this optimized molecular dynamic approach to advance the current understanding on chemical–physical interactions within polymer/surfactant mixtures at the molecular level and enable emerging engineering solutions.

## 1. Introduction

Polymers and surfactants are essential additives that have been frequently used in petroleum engineering [1,2,3,4], process intensification [5], mass transportation [6], sewage systems [7], drag delivery [8,9], etc. Within a polymer/surfactant mixture, macromolecule chains and surfactant micelles can chemically/physically interact to generate unique structures/phases such as swollen cages, bottlebrushes, etc. [10]. Those structures can bring programmable viscosity and the reduction of surface tension, which can significantly influence the downstream applications in detergents, pharmaceuticals, and cosmetics [11]. Some studies have been performed to understand the mechanism for those chemically/physically interactions in mixtures without salt [12,13,14,15,16], however, the understanding of complex systems containing salt remains to be exploited, where salt usually plays a critical role to stabilize the system as counter-ions. 

The inner structures of the aggregated phase and the chemical–physical interactions between polymers and surfactants have been extensively investigated to reveal the rheological behaviors for a multi-phase mixed solution. Researchers have developed instrumental approaches including viscometry [17], light scattering [16,18,19,20,21,22,23,24,25], Nuclear Magnetic Resonance (NMR) spectroscopy [26,27,28,29,30,31], Small Angle X-ray Scattering (SAXS) measurements [32,33], Small Angle Neutron Scattering (SANS) [31], Scanning Electron Microscopy (SEM) [8,34], etc. to directly unveil the structures and the interaction between polymers and surfactants down to a length scale of nanometers. Nevertheless, these instrumental techniques cannot provide answers at the single molecule level due to the instrumental limitations.

Molecular dynamics (MD) simulation, a powerful tool to study microstructures in chemistry, materials, biology, and other subjects [35,36,37,38,39], offers a low-cost and precision solution to understand the chemical–physical interactions and associated rheological behaviors compared to the instrumental approaches. One drawback for MD simulation is the time-consumption caused by the significant computation load. In recent decades, the coarse-grained (CG) model, a mathematical optimization model in MD simulation, has shown its superiority in significantly reducing the load of computation while ensuring the accuracy to a great extent when being used to study the viscosity of fluids [40,41,42,43]. Marrink developed a parameterized force field for lipids and surfactants, which has been extended to study different colloidal/polymer/surfactant systems (e.g., polystyrene (PS)/polyethylene oxide (PEO) systems and sodium dodecyl sulfate (SDS)/PAM systems [44,45,46]). Therefore, the combination of the CG model and MARTINI force field can be an effective method to study the rheological behaviors in polymer/surfactant systems. 

This project proposes a new approach to explore the chemical–physical interactions in complex system based on our works on using the MD computational method [47,48]. Specifically, we utilized Coarse-Grained Nonequilibrium (CG-NEMD) simulation to study the physical–chemical interactions, phase transitions, and associated rheological characteristics in the PAM/CTAC/Salt multiphase aqueous system. The simulations were performed to analytically and systematically investigate the shear viscosity of the system by varying the shear rates, timescale, and the concentrations of the components. We aimed to reveal the inner structures and interactions within the PAM/CTAC/Salt complex and explore the mechanism of associated changes in rheological behaviors.

## 2. Computational Methods

### 2.1. Interactions

The interactions were defined with a MARTINI coarse-grained model (CGM) [45]. Four interactions were considered: polar (P), nonpolar (N), apolar (C), and charged (Q). Each type has a subtype default set by the model to distinguish the levels of interactions. “S” is a special type representing the ring structure.

As shown in Figure 1, CTA^+^ is represented by four apolar beads (the hydrophobic tail) and a charged bead (Q_0_, hydrophilic head). The hydrophobic tail is described by two types of apolar (C) sites with varied polarity influenced by the chemical structure. Sal^−^ is represented by a charged bead Q_a_ and a triangular ring consisting of three SC_4_ beads. Na^+^ and Cl^−^ are described by Q_d_ and Q_a_, respectively. The PAM molecule consists of alternating SC_1_ and P_5_ particles [13,49] with the SC_1_ beads representing the carbon backbone and P_5_ beads representing the amide groups. The polymerization degree for PAM was pre-set as n = 50. We used particle P_4_ to represent the general water in the system, according to the MARTINI force field. To avoid freezing, we set 10% of water as an antifreeze agent (BP_4_).

### 2.2. Bonded Interaction and Nonbonded Interaction

Bond interaction between two neighboring beads is determined by a harmonic potential [45]
(1)$$U_{bond}(r_{ij})=\frac{1}{2}K_{bond}{\left(r_{ij}-r_{0}\right)}^{2}$$
where $K_{\mathrm{bond}}$ and $r_{0}$ represent the spring constant and the equilibrium distance, respectively. For CTA^+^, the spring constant is 1250 KJ·mol^−1^·nm^−2^ and the equilibrium distance is 0.47 nm. The equilibrium distance for the ring of Sal^−^ and PAM are 0.27 nm and 0.28 nm, respectively.

The stiffness of a chain is described by a weak harmonic potential [45]:(2)$$U_{angle}(\theta )=\frac{1}{2}K_{angle}{\left(\cos\theta -\cos\theta_{0}\right)}^{2}$$
where $K_{\mathrm{angle}}$ represents the force constant and $\theta_{0}$ represents the equilibrium bond angle. The force constant and equilibrium bond angle are *K*_angle_ = 25 KJ·mol^−1^and $\theta_{0}=180^{\circ }$ for aliphatic chains. 

The nonbonded sites *i* and *j* interact through a Lennard–Jones potential [45]:(3)$$U_{LJ}(r_{ij})=4\varepsilon_{ij}\left[{\left(\frac{\sigma_{ij}}{r_{ij}}\right)}^{12}-{\left(\frac{\sigma_{ij}}{r_{ij}}\right)}^{6}\right]$$
where $\sigma_{\mathrm{ij}}$ represents the effective distance of approach between two beads and $\varepsilon_{ij}$ is the depth of the potential well. The potential is truncated and shifted at a cutoff distance r=1.2 nm, which can be found in many studies [50,51,52]. $\sigma_{\mathrm{ij}}$ and $\varepsilon_{ij}$ are defined by the MARTINI force field [45]. 

The electrostatic interaction between the charged beads *i* and *j* is defined as [45]:(4)$$U_{el}(r_{ij})=\frac{q_{i}q_{j}}{4\pi \varepsilon_{0}\varepsilon_{r}r_{ij}}$$
where $q_{i}$ and $q_{j}$ are the charges; $\varepsilon_{0}$ and $\varepsilon_{r}$ are the permittivity of the vacuum and the relative dielectric constant, respectively; and $\varepsilon_{r}$ is 15 [45]. 

### 2.3. Simulations

The Large-scale Atomic/Molecular Massively Parallel Simulator(LAMMPS) package was used to calculate the shear viscosity of the salt-added polymer and surfactant solutions. PACKMOL software was used to obtain a random distribution of surfactants, salt, polymers, and water particles in a simulation cube with dimensions of 23.1 nm × 23.1 nm × 23.1 nm. The van der Waals interaction shifted from 0.9 to 1.2 nm, and the electrostatic interaction shifted from 0.0 nm to 1.2 nm. The particle–particle particle-mesh (PPPM) solver was used to calculate the long-range electrostatic interactions. Minimization, isothermal-isobaric ensemble(NPT) simulation, and Canonical ensemble(NVT) simulation were carried out at a temperature of 300 K via a Nose-Hoover thermostat and under a pressure of 1 bar prior to calculating the shear viscosity. The timestep was 20 fs and the NVT simulation lasted for 600 ns to reach equilibrium. The simulation time was sufficient for the system to get to an equilibrium state because the potential energy will not change after 400 ns. The potential energy curve as a function of time can be found in Appendix A. The shear viscosity was calculated from NEMD simulation using the equilibrium data by deforming the box. This means that we needed to obtain the equilibrium structure of the simulation system before calculating the viscosity. Then, the viscosity was calculated by a non-equilibrium MD (NEMD) simulation by changing the shape of the simulation box. More simulation details can be found in the Appendix A.

## 3. Results and Discussion

In this study, the NEMD simulation was employed to understand the interactions in the PAM and CTAC mixture, and the associate impact on the rheology behavior of the PAM and CTAC containing aqueous solutions with added salt (PAM/CTAC/Salt multiphase aqueous solutions). In our simulation, there were two series of PAM/CTAC/Salt multiphase aqueous solutions. One was the PAM based solution, which had a given concentration of PAM, but variable concentrations of CTAC from 0.1 to 0.3 mol·L^−1^. Another was the CTAC based solution with a fixed CTAC concentration, but variable concentrations of PAM from 2.96 × 10^−3^ to 2.02 × 10^−2^ mol·L^−1^. NaSal is used as a counter-ion salt to form stable micelle structures with CTAC. In this work, the mole ratio of NaSal to CTAC (R) was 0.8 in both PAM based solutions and CTAC based solutions. Surfactants can self-assembly into micelles with various shapes such as spherical, wormlike, and branched micelles. As mentioned in the introduction, surfactants are efficient in drag reduction. The reason why surfactants can be used as drag reducing additives is that surfactant micelles can form shear induced structures (SIS). When the shear induced structures are destroyed, the solution will lose its drag reduction efficiency. The change in micelle structures is also called shear induced phase transition (SPS) [53,54]. The SPS phenomenon also leads to the shear thickening phenomenon in rheology behavior, which means that the evolution of structures can influence the rheology behavior during the shear process. In our study, changes in the concentrations of CTAC and/or PAM in the mixed solutions could influence the initial structures of the mixtures and further influence the rheology behavior of the solutions. Thus, the structure evolution (phase transition) and the rheology behavior can be studied. 

### 3.1. Study of Shear Viscosity for PAM/CTAC/Salt Multiphase Aqueous System

The shear viscosities of the PAM/CTAC/Salt multiphase aqueous were analyzed at shearing rates ranging from 7.79 × 10^8^ to 1.73 × 10^11^ s^−1^. The viscosity results of the PAM based solutions in Figure 2a show an initial plateau under low shear rates (from 7.79 × 10^8^ to 1.3 × 10^9^ s^−1^). When the shear rates increase from 1.3 × 10^9^ to 1.73 × 10^11^ s^−1^, all samples showed a typical shear thinning phenomenon. Thus, we think that the point when the shear rate is 1.3 × 10^9^ s^−1^ is considered to be the turning point; before this point is the plateau zone, and after this point is the shear thinning zone. The steady state shear viscosity data agreed well with the reported experimental results of the pure surfactant solutions and pure polymer solutions because both results had the plateau and the shear thinning zone [55,56,57]. The shear viscosity data of the CTAC based solutions in Figure 2b also show an initial plateau at low shear rates and shear thinning in the high shear rates region. The above results indicate that even with various concentrations of CTAC or PAM, the trend of the change in the shear viscosity was similar in the PAM/CTAC/Salt multiphase aqueous system. 

We have to point out that the shear rates used in our simulation were much larger than those applied in the experiments. This is because the calculation area is small in molecular simulation, and the shear rates applied in the experiments were not large enough to access reliable data in the molecular simulations. If the shear rate is not enough, the calculation noise will be very large and the statistical data cannot be accepted because the viscosity value will become a negative number, which has no physical significance. It can be found in the published studies [47,50,52,58] that the shear rates in the molecular simulation were far above the experimental data, but the simulation results shown qualitatively agreed with experimental results because the simulation curves and the experimental curves had similar trends (both the plateau and the shear thinning zone). However, if the shear rate is too large, the random motion of the sites will also lead to no physical significance viscosity value because the data cannot be well analyzed. Thus, the reliable shear rates have a range for each simulation case, for our simulation, the reliable range was from 7.79 × 10^8^ to 1.73 × 10^11^ s^−1^ and the large shear rates make sense. The corresponding inner structures can give us valuable information in understanding the rheology behavior of polymer/surfactant solutions. 

### 3.2. Time Dependent Shear Viscosity and Structure Evolution for PAM/CTAC/Salt Multiphase Aqueous System

#### 3.2.1. Medium and High Shear Rates 

The viscosity values of the PAM/CTAC/Salt solutions, as a function of simulation time, are shown in Figure 3a. The data of the PAM based solutions revealed a similar trend at a medium shear rate (1.3 × 10^10^ s^−1^) and a high (8.65 × 10^10^ s^−1^) shear rate. Herein, the data can be regarded as an equilibrium value when the value of the viscosity fluctuates ≤2% and last for a simulation time of ≥2 ns. It can be observed in Figure 3a that there was a set of data points at the beginning of the simulation, followed by a steady region at a later stage of simulation. The viscosity tended to reach its stable region earlier at the high shear rate than under the medium shear rate. More information can be found in Appendix A
Appendix A.

Figure 3b,c show the structure of the mixture under medium shear rates (1.3× 10^10^ s^−^^1^ in Figure 3a) and high shear rates (8.65 × 10^10^ s^−^^1^ in Figure 3a), respectively. It can be seen that the CTAC micelles and PAM polymer chains orientate along the flow direction under the shear force and form a stretched band at medium and high shear rates. The alignments of the mixture along the flow direction could lead to the reduction in viscosity of the mixture solution, and the saturated viscosity value whilst the phase interactions became stable.

#### 3.2.2. Low Shear Rates for PAM Based Solutions

The volume fraction of CTAC was larger than that of PAM in the PAM based solutions, which means that the rheological behavior of the mixture will be significantly influenced by surfactants, especially under low shear rates. As shown in Figure 4a, typical “two peaks” curves were obtained with a CTAC concentration of 0.3 mol·L^−1^ (more curves are available in the Appendix A
Appendix A), which has been previously reported in many surfactant solution experiments [59,60]. After the two peaks, the viscosity curve finally reached a steady state after 9 ns, and the plateau lasted until 11 ns, which can be easily noted in Figure 4a. Next, the time dependent viscosities of the PAM based solutions under low shear rates are shown in the small figure in Figure 4a, and the time dependent viscosities of the PAM based solutions under low shear rates are shown in Figure 4a. Figure 4b shows a multi-stage evolution for the viscosity and is a theoretical construct summarization of Figure 4a. The multi-stage evolution can be approximately divided into five stages: oscillating shear-thickening (Stage I), shear-thinning (Stage II), oscillating sharp shear-thickening (Stage III), oscillating adjustment (Stage IV), and oscillating plateau (Stage V). 

To further understand the relationship between the structural evolution and the changes in shear viscosity, the continuous phase changes for the mixed solution in Figure 4a were analyzed at a shear rate of 1.08 × 10^9^ s^−1^. A series of snapshots (Figure 5a–g) were taken from the CGMD simulation to visualize the time dependent multi-stage phase evolution. In Figure 4b, from a to b, there was an increase in viscosity, but the increase only lasted for a short time. From Stage I in Figure 5a,b, we can see that the structure of the mixture had no significant changes, apart from the slight stretching under shearing (top view). The oscillating increase in viscosity at Stage I can be attributed to the increase of viscosity due to the extension of the PAM molecules [61,62] and the elastic response of the resistance of disentanglement action of surfactant micelles [59]. At Stage II, the viscosity decreased with time and the stage lasted for about 0.8 ns (Figure 4b, from b to c). From the corresponding snapshots (Figure 5b,c from the left view), we observed that the structures were slightly stretched, and the structures became smaller, indicating the alignment of structures with flow direction, which is likely to induce a slight decrease in the viscosity [52].

In the following stage (Figure 4b, from c to d), the viscosity increased with time and this part lasted for about 1 ns. In this stage, we can see from the snapshots in Figure 5 that the interconnected networks consisting of PAM and CTAC were very stretched, both the CTAC micelles and PAM polymer chains were elongated, and the original tight structures were unfolded (Figure 5c,d). The expenditure of the contact area with fluids leads to an increase in viscosity [63]. Coupled with the alignment of structures to the shear direction (from the left view in Figure 5c,d), the viscosity showed an oscillation increase. Next, a reduction in viscosity appeared at Stage IV (Figure 4b, from d to e), where the reduction lasted for a long time, about 7 ns. The viscosity decreased very slowly with time. We believe that this reduction was due to the breakage of aggregates and the further alignment of structures. In the top view images in Figure 5d,e, we observed some micelles of small size as well as a reduction in the projection area of the aggregate structures. In Stage V, the shear viscosity reached a constant value independent of the time duration (Figure 4, from f to g), the viscosity reached a plateau regime at 9 ns, and the stage lasted for 2 ns. From Figure 5f to 5g, this plateau can be attributed to the dynamical equilibrium between inner structures (Figure 5f,g).

#### 3.2.3. Low Shear Rates for CTAC Based Solutions

Compared with the PAM based solutions, the viscosity curves (Figure 6a) of the CTAC based solutions showed less complexity with only one peak in Figure 6b. The curve in Figure 6b is a theoretical construct tendency for the viscosity curves of CTAC based solutions under low shear rates (Figure 6a). The volume fraction of the PAM in the CTAC based solutions was much larger than that in the PAM based solutions due to the increase in the concentration of PAM in the CTAC based solutions while the volume fraction of the PAM was close to or larger than that of the CTAC micelles. Thus, PAM becomes the main factor that influences the rheology of the CTAC based solutions.

The structure evolution for the mixture at a shear rate of 8.65 × 10^8^ s^−1^ from Figure 6a was captured and presented in Figure 7 to analyze the time dependent phase evolution within the mixture. The CTAC micelles exist as a minority phase, which is mostly wrapped by PAM polymer chains. When shear force is applied, the PAM polymer chains first endure most of the shear load and then extend, leading to the increase in viscosity (Figure 6b, from a to b). This stage only lasts for a short time when compared to the other two stages. The resistance of the disentanglement of CTAC micelles also contributes to the increment in viscosity. This can be traced from the top view in Figure 7a,b. Then, the aggregates, especially CTAC micelles, gradually align with the flow (Figure 7b–d), resulting in a continuous decrease in both viscosity and stabilization (Figure 6b, from b to c). At the last stage, the viscosity value reached a plateau regime (Figure 6b, from c to d); from the snapshots in Figure 7c,d, it can be observed that the structures have reached a dynamic equilibrium and no structural changes appear. 

In the CTAC based solutions, surfactant micelles are the minority and wrapped by PAM polymer chains (PAM molecules) in the PAM-dominated solution, while the PAM polymer chains are the backbones of the aggregates. The excess PAM polymer chains are connected by insufficient surfactant micelles, and PAM chains cannot form strong network structures by themselves. When the aggregates are stretched and unfolded, the structures are divergent and almost have no capability in improving the fluid retention capacity of the solution, and the viscosity no longer increases. Comparing the CTAC based solution and PAM based solution, it can be found that the viscosity is more influenced by the structural evolution of the micelles and the viscosity curves are complicated when CTAC plays a dominant role in rheology. When the concentration of PAM increases, the curves become much smoother due to the long chains of PAM. The polymer chains can absorb and transfer the shear stress to the elastic energy during an enduring and gradual process [47].

### 3.3. Relation between Shear Rates and Characteristic Time

After calculating the shear viscosity versus time, we now compare the relation between shear rates and characteristic time. As some of the simulation results data were fluctuant, where it is difficult to calculate the characteristic time; here, we selected one group of data as an example. We used the “plateau” time as the characteristic time to describe the dynamics of the solutions, which has been used in many other studies [64,65]. The plateau time *t* is defined at which viscosity reaches the dynamically stable value in the plateau region under the shear rate. It has been found that the relation between characteristic time and shear rate is consistent with the power low dependence of the form *t* ∝ γ*^n^* in experiments [64,65]. Here are our simulation results.

We chose a solution with C_CTAC_ = 0.3 mol·L^−1^ and C_PAM_ = 2.96 × 10^−3^ mol·L^−1^ as an example. 

Under low shear rates (shear rates equal and smaller than 1.3 × 10^9^ s^−1^, Figure 8a), the four curves were similar to each other, thus we used one plateau time *t*_1_ to represent the four curves, and the corresponding shear rate was 1.3 × 10^9^ s^−1^, which was the turning point (shear thinning starts). Similarly, four other characteristic time points can be obtained from Figure 8b. When the shear rate is 1.73 × 10^11^ s^−1^, it is hard to distinguish the plateau time point, thus we did not use this data. Figure 9 shows the relation between the characteristic time and the shear rates. 

From the figure, we can note that the relation between characteristic time and shear rate is consistent with the power law dependence of the form *t* ∝ γ*^n^*, which coincides with the experimental studies [65,66]. The exponent n is −0.94, which shows an agreement with Ma’s experimental results [65]. 

### 3.4. Shear Viscosity Versus the Concentration of CTAC for Polymer/Surfactant Solutions

We next summarize the shear viscosity as a function of CTAC concentration for all PAM based solutions in Appendix A
Appendix A. The results showed that with the increase in the concentration of CTAC, the shear viscosity increased and then decreased, indicating good agreement with the reported experimental results [67,68,69]. Here, we analyzed the influence of CTAC concentration in a mixture solution by using the initial equilibrated morphology data (before shear action). According to Wang’s published work [70], the shapes of the micelles can be classified into three certain types based on the length ratio of the long axis to the short axis of the micelle. In the present work, when the ratio was around 1, the micelle was defined as a spherical micelle; when the ratio was larger than 1 and there was no branch in the micelle, the micelle was defined as a wormlike micelle; when a branch exists, the micelle was a branched micelle. Figure 10 shows the representative morphologies of the micelle and branch node. The representative morphology was selected from the simulation results. From the figure, the spherical micelle had the smallest size; two or more spherical micelles coalesced and then formed a wormlike micelle; two wormlike micelles or one spherical micelle and one wormlike micelle coalesced and then a branch node exists and branched micelle forms, the branched micelle had the largest size [51]. Thus, if the concentration of CTAC is low, only small sized spherical micelles and wormlike micelles form; if the concentration increases, large wormlike micelles, and branched micelles will form. The concentration will influence the morphology of the micelle, which can further lead to different viscosity behaviors in rheology. Figure 11 shows the morphology proportion, number of branch nodes, and viscosity as a function of CTAC concentration in PAM based solutions. 

In our simulation, the volume fraction of PAM was lower than that of CTAC in the PAM based solutions, thus the viscosity of the mixture is likely to be affected by the rheology property of CTAC. In addition, the CTAC micelles have a diverse morphological adaptivity and can self-assemble into spherical micelles, wormlike micelles, branched micelles, network micelles, and so on. Those morphological changes also have a large impact on the viscosity of solution.

It can be seen in Figure 11 that CTAC micelles show a continuous morphological transition among the spherical shape, wormlike shape, and branched shape when the concentration of CTAC increased (more morphology images are in Appendix A
Appendix A). This transition can explain the viscosity changes in Figure 11. From Figure 11, we can observe that with the increase in the concentration of CTAC, CTA^+^ ions can form larger sized micelles such as wormlike micelles and branched micelles, and small size micelles (spherical micelles) disappear. Furthermore, the larger micelle proportion also increased with the increase in CTAC concentration. In detail, when the CTAC concentration is from 0.1 to 0.2 mol·L^−1^, only spherical micelles and wormlike micelles exist, and the proportion of wormlike micelles increases with the increase in CTAC concentration, which indicates that the micelle size increases with the increase in CTAC concentration. Since the wormlike micelles can be more effective in resisting the shear action and suppress the fluctuation of the flow field than spherical ones, so the viscosity increases with the increase in proportion of wormlike micelle. However, branched micelles appear when the CTAC concentration further increases. Although a high CTAC concentration leads to the formation of large sized micelles (branched micelles and wormlike micelles), the formation of branched micelles can reduce the viscosity of the solutions as branches provide a sliding mechanism for stress relaxation, as has been reported [50,70,71], leading to a reduction in the viscosity of the solutions. When the CTAC concentration is further increased from 0.25 to 0.3 mol·L^−1^, both the branch nodes and the proportion of branched micelles increase. The stress relaxation is faster in a solution with a higher degree of branching [71], thus the viscosity has a further decrease. It should be noted that the concentration of micelles also increases during the period, but it is still insufficient to compensate the reduction effect on viscosity induced by the morphological change of CTAC micelles.

### 3.5. Shear Viscosity Versus the Concentration of PAM for Polymer/Surfactant Solutions

The shear viscosities of CTAC based solutions are summarized as a function of PAM concentration in Appendix A
Appendix A, and the corresponding structure snapshots are in Appendix A. The group at a shear rate of 8.65 × 10^8^ was selected for analyzing the influence of PAM concentration in a mixture solution (Figure 12). The initial equilibrated morphology data (before shear action) were used to analyze the micelle shape and the morphology proportion. With the increase in PAM concentration, the shear viscosity of solutions first decreased and then increased. In the polymer/surfactant solutions, the PAM polymer chains normally work as “bridges” and bind surfactant micelles together [47]. When we increased the concentration of PAM from 2.96 × 10^−3^ to 4.45 × 10^−3^ mol·L^−1^, the interaction between the polymers and surfactants became stronger and most of the micelles changed from wormlike micelles to branched micelles. The existence of branched micelles led to the reduction in viscosity for the mixed solutions, thus the viscosity of the CTAC based solutions first decreased with the increase in concentration of PAM.

The viscosity then increased slightly due to the further increase in the concentration of PAM (which increased to 7.41 × 10^−3^ mol·L^−1^), but was still a relatively low value due to the continuous formation of branched micelles in the CTAC based solution. When the concentration of PAM was higher than 7.41 × 10^−3^ mol·L^−1^, the volume fraction of PAM was larger than that of CTAC, which means that polymers dominate the rheology behavior and the polymer chains become the framework of the mixed structures. Thus, the morphology of the surfactant micelles is limited due to the polymer framework. Branched micelles disappear and the size of the micelles decreases, and small wormlike micelles form instead. The disappearance of branched micelles leads to the increase of the viscosity. Meanwhile, the increase in the concentration of PAM also results in the increase in viscosity. The viscosity change caused by PAM can compensate for the reduction effect on viscosity induced by the morphological change of CTAC micelles from wormlike micelles to spherical micelles, and leads to the further increase in viscosity.

## 4. Conclusions

We used a NEMD based numerical approach to understand the chemical-physical interactions within multi-phase PAM/CTAC/NaSal solutions and their impacts on macroscopic rheological behaviors. By analyzing the relationship between shear viscosity and shear rates, a plateau followed by a shear thinning zone was observed, which is in good agreement with the experimental results. We also discovered the “two peaks” phenomenon for the time by simulation after investigating the time dependent shear viscosity and the corresponding inner structural evolution between different phases under different shear rates in the PAM based solutions. The viscosity curve of the CTAC based multi-phase solution became smoother than that of the PAM based (CTAC dominated) solutions due to the fact that PAM chains absorb and transfer the shear energy into elastic energy by extending/stretching the polymer chains during an enduring and gradual process. The relation between characteristic time and shear rate is consistent with the power law. After looking into the relationship between shear viscosity and the concentration of the solutes, an explicit reduction in viscosity was found because of the phase transition of CTAC micelles from spherical micelles into branched micelles.

## Figures and Tables

**Figure 1 polymers-12-00265-f001:**
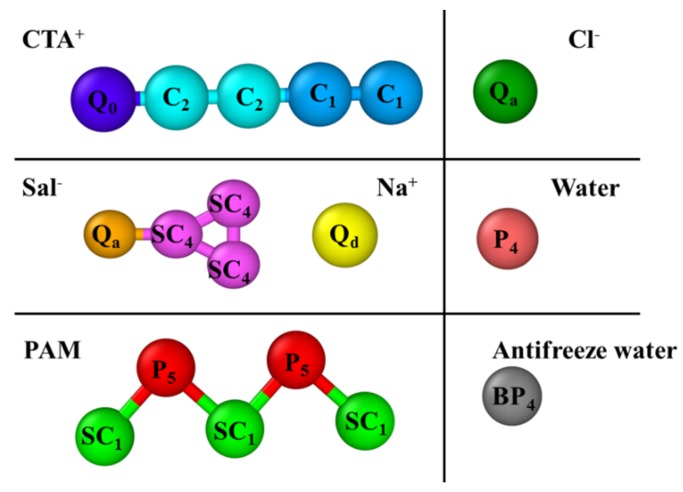
Mapping of cetyltrimethyl ammonium chloride (CTAC), polyacrylamide (PAM), sodium salicylate (NaSal) water, and antifreeze water in the MARTINI coarse-grained model.

**Figure 2 polymers-12-00265-f002:**
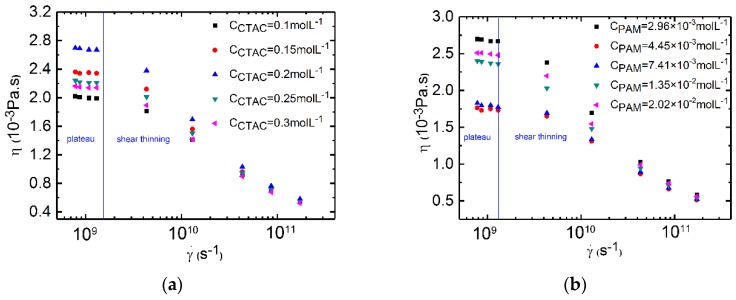
(**a**) Simulation of shear viscosity versus shear rates for the PAM based solutions. C_PAM_ = 2.96 × 10^−3^ mol·L^−1^, Counter ion salt: NaSal, R = 0.8, T = 300 K. Black square: C_CTAC_ = 0.1 mol·L^−1^. Red circle: C_CTAC_ = 0.15 mol·L^−1^. Blue triangle: C_CTAC_ = 0.2 mol·L^−1^. Dark cyan triangle: C_CTAC_ = 0.25 mol·L^−1^. Magenta triangle: C_CTAC_ = 0.3 mol·L^−1^. (**b**) Simulation of shear viscosity versus shear rates for CTAC based solutions. C_CTAC_ = 0.2 mol·L^−1^, counter ion salt: NaSal, R = 0.8, T = 300 K. Black square: C_PAM_ = 2.96 × 10^−3^ mol·L^−1^. Red circle: C_PAM_ = 4.45 × 10^−3^ mol·L^−1^. Blue triangle: C_PAM_ = 7.41 × 10^−3^ mol·L^−1^. Dark cyan triangle: C_PAM_ = 1.35 × 10^−2^ mol·L^−1^. Magenta triangle: C_PAM_ = 2.02 × 10^−2^ mol·L^−1^.

**Figure 3 polymers-12-00265-f003:**
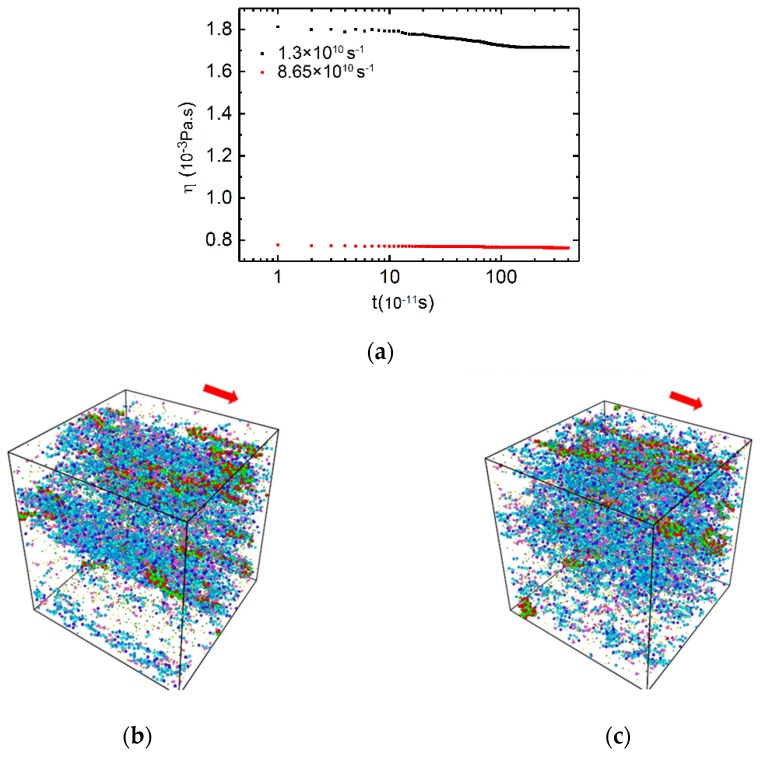
(**a**) Shear viscosity versus time at medium and high shear rates for mixtures shown in logarithmic axis. C_CTAC_ = 0.2 mol·L^−1^, C_PAM_ = 2.96 × 10^−3^ mol·L^−1^, counter ion salt is NaSal, R is 0.8, T = 300 K. Black square: shear rate is 1.3 × 10^10^ s^−1^. Red circle: shear rate is 8.65 × 10^10^ s^−1^. (**b**) Snapshot of the PAM based solution at 4 ns at a shear rate of 1.3 × 10^10^ s^−1^. (**c**) Snapshot of the PAM based solution at 4.4 ns at a shear rate of 8.65 × 10^10^ s^−1^. Color scheme: purple, hydrophilic part of CTAC; cyan and blue, hydrophobic tail of CTAC; magenta, aromatic ring of Sal^-^; khaki, charge group of Sal^−^; yellow, Na^+^; dark green, Cl^−^; red, the acidammide group of PAM; green, carbon backbone of PAM. Water sites (reddish) and antifreeze water sites (grey) were hidden for clear display. The applied flow direction is indicated by a red arrow in (b) and (c).

**Figure 4 polymers-12-00265-f004:**
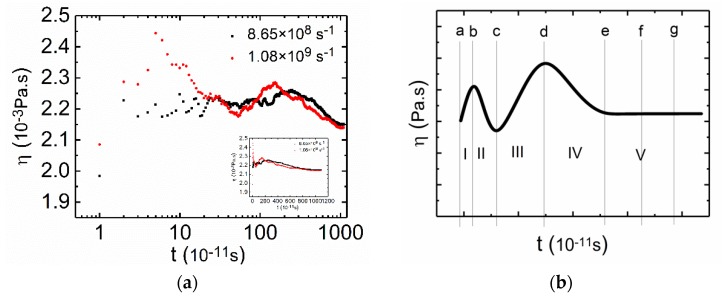
(**a**) Shear viscosity versus time at low rates for the PAM based solutions shown in the logarithmic axis. The small figure is the same curve shown in linear axis. C_CTAC_ = 0.3 mol·L^−1^, C_PAM_ = 2.96 × 10^−3^ mol·L^−1^, counter ion salt is NaSal, R is 0.8, T = 300 K. Black square: shear rate is 8.65 × 10^8^ s^−1^. Red circle: shear rate is 1.08 × 10^9^ s^−1^. (**b**) Schematic curve of the PAM based solution under low shear rates.

**Figure 5 polymers-12-00265-f005:**
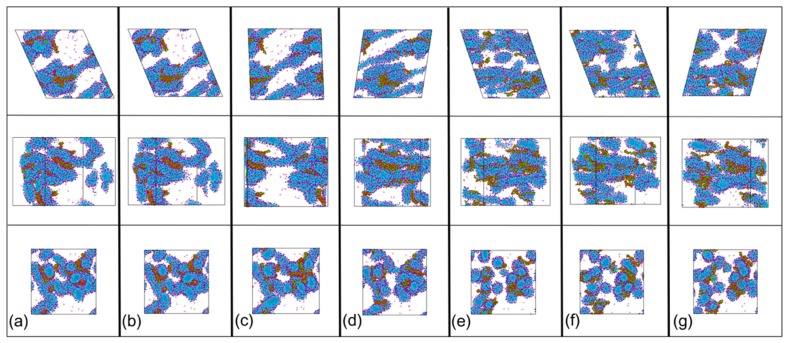
Snapshot of the PAM based solution when the shear rate is 1.08 × 10^9^ s^−1^. (**a**) time: 0.5 ns; (**b**) time: 0.55 ns; (**c**) time: 0.89 ns; (**d**) time: 1.99 ns; (**e**) time: 8 ns; (**f**) time: 9.8 ns; (**g**) time: 10.4 ns. Top view: first line, front view: second line, left view: third line. Color scheme: purple, hydrophilic part of CTAC; cyan and blue, hydrophobic tail of CTAC; magenta, aromatic ring of Sal^−^; khaki, charge group of Sal^−^; red, the acidammide group of PAM; green, carbon backbone of PAM. Na^+^ (yellow), Cl^−^ (dark green), water sites (reddish), and antifreeze water sites (grey) were hidden.

**Figure 6 polymers-12-00265-f006:**
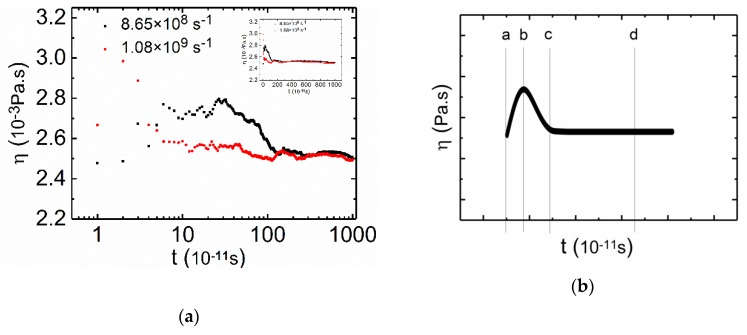
(**a**) Shear viscosity versus time at low shear rates for the CTAC based solutions shown in the logarithmic axis. The small figure is the same curve shown in linear axis. C_CTAC_ = 0.2 mol·L^−1^, C_PAM_ = 2.02 × 10^−2^ mol·L^−1^, counter ion salt is NaSal, R is 0.8, T = 300 K. Black square: shear rate is 8.65 × 10^8^ s^−1^. Red circle: shear rate is 1.08 × 10^9^ s^−1^. (**b**) Schematic diagram of the CTAC based solution under low shear rates.

**Figure 7 polymers-12-00265-f007:**
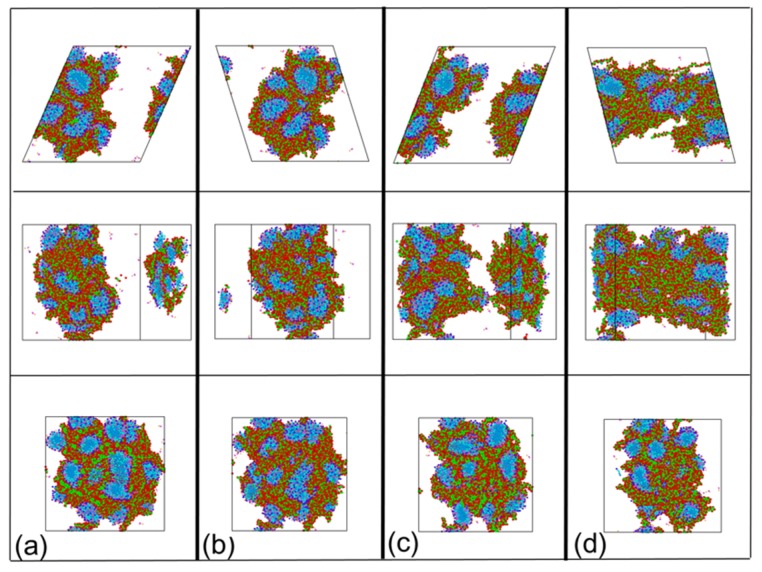
Snapshot of the CTAC based solution when the shear rate is 8.65 × 10^8^ s^−1^. (**a**) time: 0.5 ns; (**b**) time: 0.8 ns; (**c**) time: 1.6 ns; (**d**) time: 5.6 ns. Top view: first line, front view: second line, left view: third line. Color scheme: purple, hydrophilic part of CTAC; cyan and blue, hydrophobic tail of CTAC; magenta, aromatic ring of Sal^−^; khaki, charge group of Sal^−^; red, the acidammide group of PAM; green, carbon backbone of PAM. Na^+^ (yellow), Cl^−^ (dark green), water sites (reddish), and antifreeze water sites (grey) were hidden for clear display.

**Figure 8 polymers-12-00265-f008:**
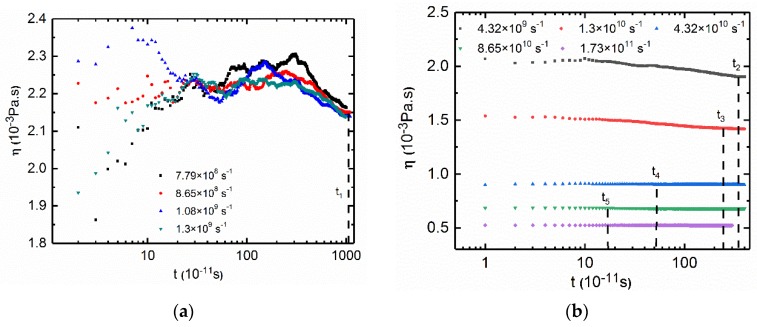
The “plateau” time of different shear rates. (**a**) “plateau” time of low shear rates; (**b**) “plateau” time of medium and high shear rates. C_CTAC_ = 0.3 mol·L^−1^, C_PAM_ = 2.96 × 10^−3^ mol·L^−1^.

**Figure 9 polymers-12-00265-f009:**
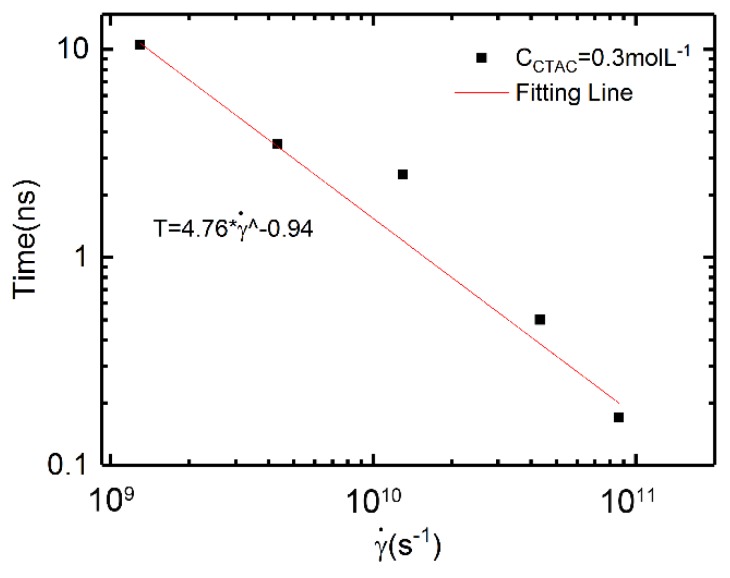
The relation between the shear rates and characteristic time.

**Figure 10 polymers-12-00265-f010:**
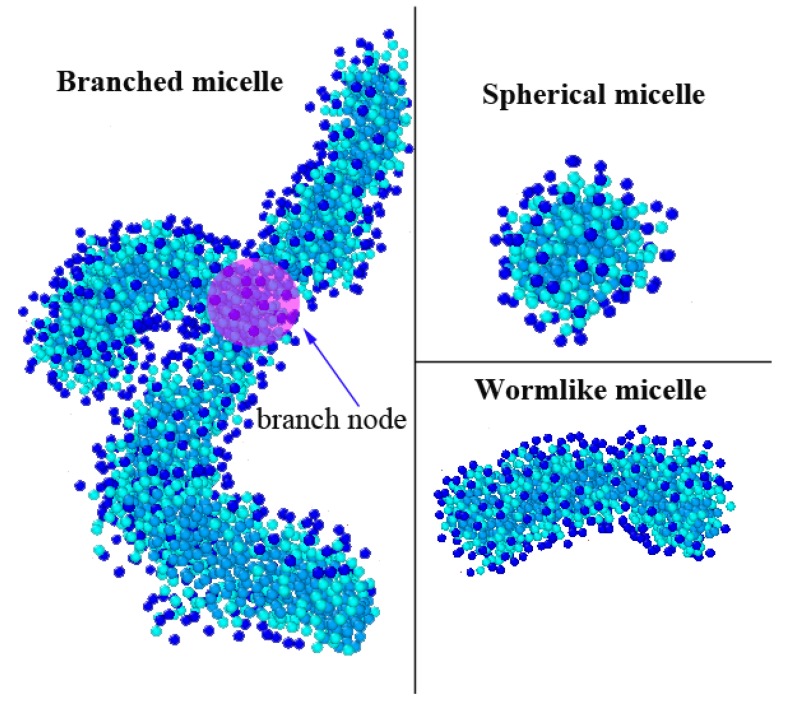
Example of a branched micelle, branch node, spherical micelle, and wormlike micelle.

**Figure 11 polymers-12-00265-f011:**
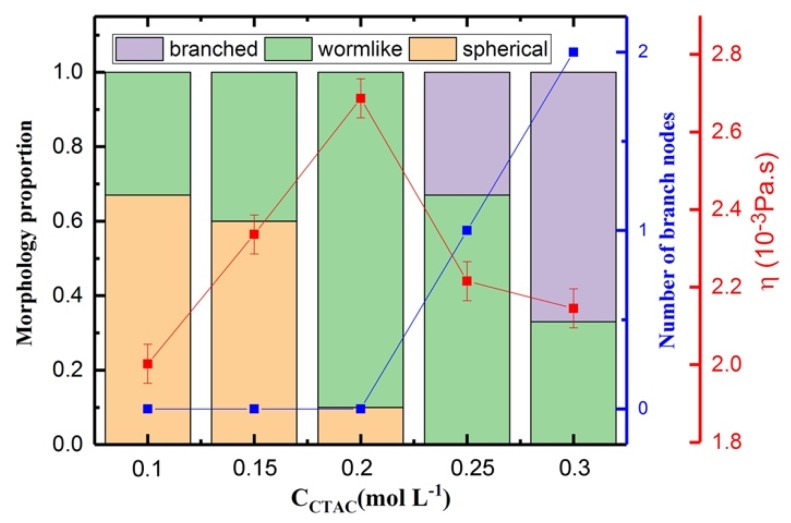
Results for the morphology proportion, number of branch nodes, and viscosity as a function of CTAC concentration.

**Figure 12 polymers-12-00265-f012:**
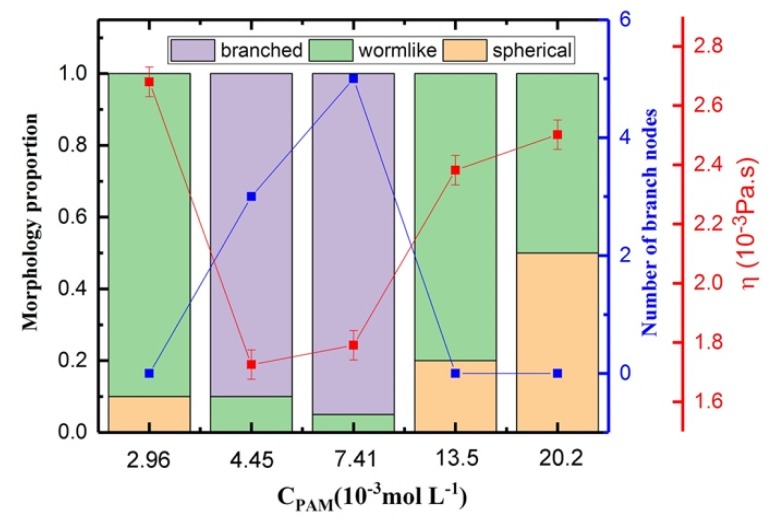
Results of the morphology proportion, number of branch nodes, and viscosity as a function of PAM concentration.

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
