# Peer review of "Interfacial Interaction Enhanced Rheological Behavior in PAM/CTAC/Salt Aqueous Solution—A Coarse-Grained Molecular Dynamics Study"

_polymers, 2020, doi:10.3390/polym12020265_

Round 1

Reviewer 1 Report

This manuscript may be published after addressing the following points.

Please discuss the origin of the shear thinning in Figure 2. What is the shear rate at which shear thinning starts? Can the authors compare it with characteristic time scales for the dynamics of the solutions? What exactly is the "qualitative agreement" between the simulations at much higher shear rates and the experiments? Please justify the high rates used in the simulations.  Why does viscosity reaches the steady-state value faster at higher shear rate?  It is hard to tell the differences between snapshot in Figure 3(b) and Figure 3(c).  In Figure 4(a), it seems that the plateau regime V has not been reached yet. Please add more detailed description of 5 regimes in Figure 4(b) and relate the viscosity change in each regime with the morphological change shown in Figure 5. Similar comment for Figure 6(b) and Figure 7.  Please add details about how different morphologies in Figure 8 are identified in the simulations. Please add separate panels showing representative morphologies in each case.  Why is there a correlation between viscosity and morphology of micelles? What is the fundamental reason?

Reviewer 2 Report

Several times it has been said that the potential "has been shifted" (e.g. page 3, line 101, and later). I have no idea what is meant by that. Is this your way to say that delta-peaks in the forces (caused by the energy truncation) is removed? please explain or reformulate.

The x-axis units in Figs. 3(a), 4 and 6 are given in a strange manner (bracket kind and size).  
